# Identifying the wide diversity of extraterrestrial purine and pyrimidine nucleobases in carbonaceous meteorites

Yasuhiro Oba [1✉], Yoshinori Takano [2], Yoshihiro Furukawa [3], Toshiki Koga[2], Daniel P. Glavin [4], Jason P. Dworkin [4] & Hiroshi Naraoka [5]

The lack of pyrimidine diversity in meteorites remains a mystery since prebiotic chemical models and laboratory experiments have predicted that these compounds can also be produced from chemical precursors found in meteorites. Here we report the detection of nucleobases in three carbonaceous meteorites using state-of-the-art analytical techniques optimized for small-scale quantification of nucleobases down to the range of parts per trillion (ppt). In addition to previously detected purine nucleobases in meteorites such as guanine and adenine, we identify various pyrimidine nucleobases such as cytosine, uracil, and thymine, and their structural isomers such as isocytosine, imidazole-4-carboxylic acid, and 6-methyluracil, respectively. Given the similarity in the molecular distribution of pyrimidines in meteorites and those in photon-processed interstellar ice analogues, some of these derivatives could have been generated by photochemical reactions prevailing in the interstellar medium and later incorporated into asteroids during solar system formation. This study demonstrates that a diversity of meteoritic nucleobases could serve as building blocks of DNA and RNA on the early Earth.

---

[1] Institute of Low Temperature Science (ILTS), Hokkaido University, N19W8, Kita-ku, Sapporo, Hokkaido 060-0189, Japan. [2] Biogeochemistry Research Center (BGC), Japan Agency for Marine-Earth Science and Technology (JAMSTEC), 2-15 Natsushima, Yokosuka, Kanagawa 237-0061, Japan. [3] Department of Earth Science, Tohoku University, 6-3 Aza-aoba, Aramaki, Aoba-ku, Sendai 980-8578, Japan. [4] Solar System Exploration Division, National Aeronautics and Space Administration (NASA), Goddard Space Flight Center (GSFC), Greenbelt, MD 20771, USA. [5] Department of Earth and Planetary Sciences, Kyushu University, 744 Motooka, Nishi-ku, Fukuoka, Fukuoka 819-0395, Japan. ✉email: oba@lowtem.hokudai.ac.jp

 1

Recent sample return missions from the asteroids Ryugu (C-type) and Bennu (B-type) led by the Japan Aerospace Exploration Agency (JAXA) and the National Aeronautics and Space Administration (NASA), respectively, will provide us with important insights into the evolution of extraterrestrial organic molecules, and potential clues regarding the origins of life on Earth through chemical analyses of pristine extraterrestrial materials that have not been significantly compromised by terrestrial contamination. The scientific justification for choosing C- and B-type asteroids for sample return missions is that these carbon-rich asteroids are plausible parent bodies of carbonaceous chondrites[1,2], in which diverse suites of primordial organic molecules of astrochemical interest, including amino acids and sugars, have been detected[3–7]. Moreover, a recent analysis of meteorite extracts using ultra-high-resolution mass spectrometry revealed the presence of several hundreds of thousands of soluble organic species in the Murchison meteorite[8–10], strongly suggesting wide chemical diversity resulting from molecular evolution in each individual thermal history of the parent body.

Nucleobases, one of the structural components of nucleic acids, have also been identified in carbonaceous chondrites (ref. [11] and references therein). There are two types of nucleobases present in DNA and RNA: pyrimidine nucleobases that consist of a single six-membered nitrogen (N) heterocyclic ring whose N atoms are always in the one and three positions and include cytosine ($C_4H_5N_3O$), uracil ($C_4H_4N_2O_2$), and thymine ($C_5H_6N_2O_2$), and purine nucleobases that consist of six-membered and five-membered two ring N-heterocyclic structures whose N atoms are always in the 1, 3, 7, and 9 positions, including adenine ($C_5H_5N_5$) and guanine ($C_5H_5N_5O$). A diverse suite of exogenous meteoritic organics, including nucleobases, may have been delivered to the early Earth during the late heavy bombardment period (~4.0–3.8 billion years ago);[12,13] therefore, the influx of such organics is considered to have played an important role in the chemical evolution of the Earth's primordial stage.

A pioneering study tracing nucleobases in carbonaceous meteorites used UV-visible spectroscopy and thin-layer chromatography to detect nucleobases extracted from 16 g of the Orgueil meteorite in the early 1960s[14], which reported the presence of N-heterocyclic molecules, including adenine and guanine, with concentrations ranging from 11–20 µg/g-meteorite or parts per million (ppm). Further detailed analyses have been conducted using more advanced methods to investigate nucleobases from other types of carbonaceous meteorites[15–19]. However, the number of nucleobases indigenous to meteorites detected to date is at most eight (seven purine bases and one pyrimidine base)[11], which is much more limited compared to the 96 different amino acids that have been identified by name in the Murchison meteorite[7]. In contrast to the limited number of nucleobases identified in meteorites, various nucleobases and related nitrogen heterocyclic molecules have been synthesized by laboratory-based simulation experiments, such as conditions simulating interstellar molecular clouds (~10 K) and meteorite parent bodies[20–25], suggesting that these classes of organic compounds are ubiquitously present in extraterrestrial environments both inside and outside the solar system.

Recent advances in analytical techniques and instruments have enabled the detection and quantification of nucleobases in mixtures of complex organic molecules such as meteorite extracts and organic residues in interstellar ice analogs, even at parts per billion levels (ppb)[19,25]. Callahan et al.[19] analyzed extracts from 12 carbonaceous meteorites and identified six purine bases, including guanine at a concentration of 20 ppb, which is three orders of magnitude lower than that reported by Hayatsu[14]. More recently, Oba et al.[25] reported a further improvement of very small-scale analyses (<100 pg per analysis) for purine and pyrimidine nucleobases using an optimized wet-chemical treatment, providing chromatographic baseline resolution for N-containing heterocyclic target molecules. Thus, we expect that the robust method described here can be applied to organic extracts from carbonaceous meteorites and returned samples using liquid chromatography coupled with high-resolution mass spectrometry[25,26].

In the present study, we analyze nucleobases extracted from the Murchison, Murray, and Tagish Lake meteorites using high-performance liquid chromatography coupled with electrospray ionization high-resolution mass spectrometry (HPLC/ESI-HRMS) to examine the molecular profiles and determine the abundance of N-containing heterocyclic molecules (Supplementary Fig. 1) in these meteorites. In addition, we analyze aqueous extracts of different Murchison meteorite specimens and soil from the Murchison meteorite fall site to determine variations in nucleobase concentrations within the same meteorite and potential sources of terrestrial contaminants.

## Results and discussion
**Purine nucleobases and related molecules in the Murchison extracts.** We detected several purine nucleobases and their analogs in both Murchison meteorite extracts (Fig. 1, Supplementary Fig. S2, and Table 1) based on their chromatographic retention time, accurate mass measurements of their parent masses, and mass fragmentation patterns in the MS/MS measurements (Fig. 1b). Guanine is typically the most abundant purine nucleobase (hereafter referred to as purine molecules, unless otherwise denoted) in all meteorites (except for Tagish Lake), followed by adenine, xanthine, and hypoxanthine as the second most abundant group, with purine, isoguanine, and 2,6-diaminopurine as the least abundant group. The total concentrations of purine molecules were 152 and 11 ppb in the aqueous extracts of two Murchison meteorite specimens (one obtained from the University of Chicago and the other from a meteorite trading company. Hereafter, each specimen is denoted as Murchison #1 and Murchison #2, respectively). Although the concentrations in the Murchison #1 extract and those reported by Callahan et al.[19] are similar, those in the Murchison #2 extract were generally one order of magnitude lower. Because the analytical conditions used for the two Murchison specimens in the present study were essentially identical, we can infer that the observed difference was due to sample heterogeneity within the Murchison meteorite (e.g., meteoritic amino acid profiles of Murchison)[8,27]. Significant heterogeneity in amino acid abundance was previously reported in different lithologies of the Tagish Lake meteorite, where total amino acid concentrations ranged from 40 ppb in the most aqueously altered samples up to 5400 ppb in the least altered lithologies[28]. The weighted average of the concentrations of purine molecules in the two Murchison specimens is shown in Table 1 and has a similar order of magnitude as that reported by Callahan et al.[19]. Note that there is a striking difference between the extraction method used in the current study and that used by Callahan et al.[19], who extracted meteorites in 95% formic acid at 100 °C for over 24 h. If such severe extraction conditions are more suitable for the extraction of nucleobases from meteorites, it is likely that the actual concentrations of nucleobases in our meteorite samples are higher than those reported here. Nevertheless, in the present study, we used a less harsh extraction method to prevent the undesirable decomposition of fragile molecules such as sugars[6] and hexamethylenetetramine[26] present in the meteorite specimens. Consequently, the weighted average concentrations in our Murchison extracts were lower than those reported previously. We concluded that the detection of purine ($C_5H_4N_4$) in the meteorite extracts strongly supported the

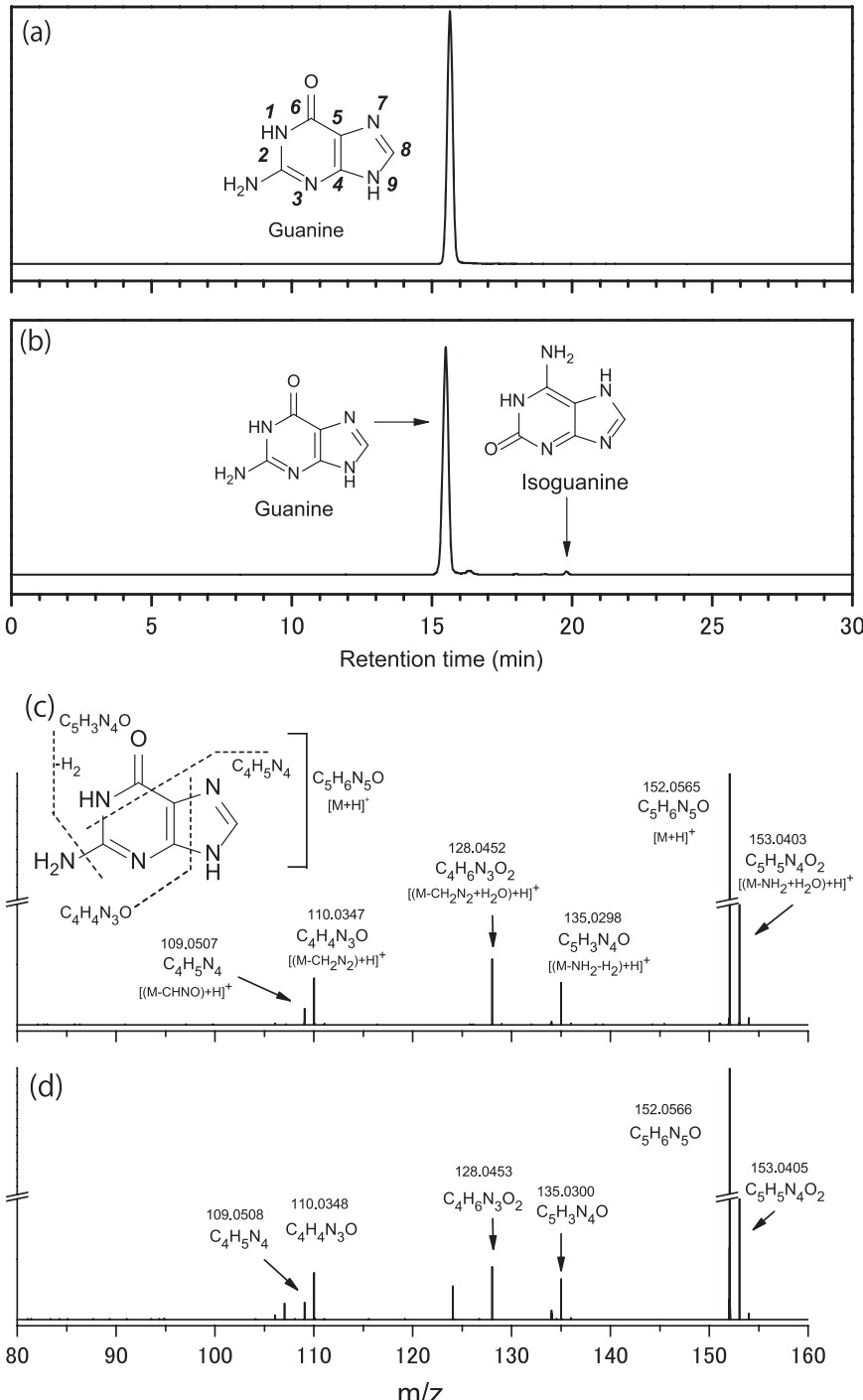

**Fig. 1 Detection of purine nucleobases.** Mass chromatograms at $m/z = 152.0567$ within a 3 ppm exact mass window at the monoisotopic mass for **a** the guanine (0.21 ng) standard reagent (STD) and **b** Murchison #1 extract were measured using an InertSustain PFP column. Additionally, isoguanine, a structural isomer of guanine, was detected in the Murchison #1 extract. Mass fragmentation patterns of **c** guanine STD and **d** guanine were detected in the Murchison #1 extract measured by MS/MS experiments. Note: "exact mass", representing a theoretical value obtained by calculation, is different from "accurate mass", which is a measured value with a high-resolution mass spectrometer. Bold italic numbers in panel **a** represent the numbering of atoms in the chemical structures of purine nucleobases.

indigenous nature of the detected purine molecules in the Murchison meteorites[19].

**Pyrimidine nucleobases in the Murchison extracts**. We detected a wide variety of pyrimidine nucleobases and their structural isomers from both Murchison extracts (Fig. 2, Supplementary Figs. 3, 4, and Table 2), most of which had not been previously

detected in meteorites. Uracil is the only pyrimidine nucleobase previously quantified in the Murchison meteorite (30 ppb in an $H_2O$ extract)[29]. Although the abundance was not quantified, uracil was also detected in the formic acid extracts of the Murchison meteorite, and the stable carbon isotopic composition ($\delta^{13}C$) was measured to be +44.5‰ (vs. VPDB standard)[18]. The uracil concentrations in the Murchison #1 and #2 meteorites were

**Table 1 Variations in the concentration (in ppb) of purine nucleobases and their analogs in the meteorite extracts and Murchison soil.**

| Name | Molecular formula | m/z (M + H⁺) | Murchison | | | | Tagish Lake | Murray | Murchison soil |
|---|---|---|---|---|---|---|---|---|---|
| | | | #1 | #2 | Weighted average | ref. [19] | | | |
| Adenine | $C_5H_5N_5$ | 136.0618 | 15 | 1 | 7 | 5 | 6 | 3 | 40 |
| Guanine | $C_5H_5N_5O$ | 152.0567 | 72 | 6 | 34 | 56 | 2 | 25 | 141 |
| Isoguanine | $C_5H_5N_5O$ | 152.0567 | 0.5 | 0.04 | 0.2 | * | n.d.** | 0.4 | n.d. |
| Xanthine | $C_5H_4N_4O_2$ | 153.0407 | 39 | 3 | 18 | 60 | 0.4 | 3 | 1 |
| Hypoxanthine | $C_5H_4N_4O$ | 137.0458 | 24 | 1 | 11 | 26 | 1 | 1 | 2 |
| Purine | $C_5H_4N_4$ | 121.0509 | 0.4 | 0.02 | 0.2 | 3 | n.d. | 0.2 | n.d. |
| 2,6-Diaminopurine | $C_5H_6N_6$ | 151.0727 | 0.2 | 0.02 | 0.1 | n.q.*** | n.d. | 0.5 | n.d. |
| | | Total | 152 | 11 | 71 | 150 | 9 | 33 | 184 |

*Blank means "not searched for" in the present study, and "not reported" in previous studies.
**n.d. not detected.
***n.q. detected, but not quantified.

15 and 1 ppb, respectively, showing that uracil is distributed heterogeneously in the Murchison meteorite. The weighted average of the uracil concentration (7 ppb) was approximately four times lower than that reported in the previous study[29], likely due to Murchison sample heterogeneity. In addition, the uracil concentration in the previous study could have been overestimated because of the co-elution of other molecules, such as carboxylic acids. As shown in Fig. 2a, in addition to uracil, we detected the structural isomers 2-imidazolecarboxylic acid and 4-imidazolecarboxylic acid in the same extract, and their concentrations exceeded that of uracil (Table 2). If the peak separation between uracil and its isomers is inadequate, as in the case shown in Supplementary Fig. 5, the uracil concentration would be overestimated. Hence, further confirmation using an appropriate LC condition (Fig. 2a) and MS/MS measurements (Fig. 2b and Supplementary Fig. 4) are required to identify a specific molecule in a mixture of complex molecules containing structural isomers.

Other pyrimidine nucleobases, such as cytosine and thymine, as well as their analogs containing a pyrimidine ring, were identified by their chromatographic retention times, accurate mass measurements of their parent masses, and mass fragmentation patterns in the MS/MS measurements (Supplementary Table 1), with concentrations ranging from 0.02 to 6 ppb (Table 2). Methylated pyrimidine nucleobases (except thymine, a methylated form of uracil), such as 5-methylcytosine, 1-methyluracil, and 6-methyluracil, are less common in biological systems than major bases such as cytosine and uracil[30–32]. The observed similar concentrations of methylated and non-methylated pyrimidine nucleobases strongly suggest that they are abiological and hence extraterrestrial in origin. Though the presence of the relatively unstable cytosine[33] is unexpected and would typically be attributed to contamination, their presence in abundances similar to isomers and methylated analogs reinforces the abiological origin of cytosine. The reason for the preservation of cytosine is unclear, perhaps it was produced at a steady state in the meteorite parent bodies and protected from deamination due to rapid drying and/or low temperatures once the hydrothermal phase on the asteroid had stopped. In addition, some coexisting species on the asteroid could have a role for the preservation of cytosine, which will be investigated in our future study.

The total concentrations of the detected pyrimidine nucleobases and their analogs (hereafter referred to as pyrimidine molecules, unless otherwise denoted) were 37 and 15 ppb in Murchison #1 and Murchison #2, respectively, and the weighted average was 24 ppb, which is ~0.1% of the total amino acid concentration in the Murchison meteorite[3]. The much lower concentrations of pyrimidine and purine molecules compared to meteoritic amino acids suggest that nucleobases are less suitable for formation in extraterrestrial environments, including meteorite parent bodies. Pyrimidine ($C_4H_4N_2$) was not identified in the two Murchison extracts, which is consistent with organic residues in UV-processed interstellar ice analogs[25]. The absence of pyrimidine in meteorite extracts suggests that the addition of side chains such as methyl (-CH₃) and amino (-NH₂) groups to pyrimidine may not be the dominant pathway to the formation of the detected pyrimidine nucleobases.

**Assessment of terrestrial contamination of nucleobases in the Murchison extract.** The analysis of extracts from the procedural blank that consisted of prebaked quartz did not yield any detectable nucleobases, indicating no terrestrial nucleobase contamination during the analytical procedures. However, we detected pyrimidine and purine nucleobases from the Murchison soil extract at concentrations ranging from 1 to 141 ppb (Tables 1 and 2), which were generally higher than those in the Murchison extracts (Tables 1 and 2). Callahan et al.[19] also detected purine and pyrimidine nucleobases in the Murchison soil formic acid extract at concentrations ranging from 7 to 1380 ppb, with cytosine being the most abundant. In contrast, except for DNA/RNA nucleobases, the Murchison soil showed less molecular diversity than the formic acid extract of the Murchison meteorite. Based on significant differences in the distributions of nucleobases and their analogs between the Murchison meteorite and soil extracts, including purine, 2,6-diaminopurine, and 6,8-diaminopurine, which were detected in the meteorite extract but not in the soil, they concluded that purines found in the Murchison meteorite are extraterrestrial in origin. We used two Murchison meteorite specimens: a 2 g specimen from the University of Chicago (Murchison #1) and a 2.7 g specimen from a meteorite trading company (Murchison #2). If terrestrial contamination dominates the meteoritic nucleobase distribution in the present study, it indicates that the meteorite size and nucleobase concentration are positively correlated. However, the actual distributions were opposite, with the concentrations of nucleobases in Murchison #1 generally being much higher than those in Murchison #2 (Tables 1 and 2); this strongly suggests that the nucleobases in the soil are not inherited by the meteorite and hence the detected nucleobases are all extraterrestrial in origin. In addition, other purine and pyrimidine molecules, such as isoguanine, 5-methylisocytosine, and purine, were not detected in the soil extracts, supporting the assumption described above. Note that alanine extracted from Murchison #1 was racemic[34], which strongly suggests that the detected alanine is extraterrestrial in origin with no evidence of significant biological contamination of the meteorite

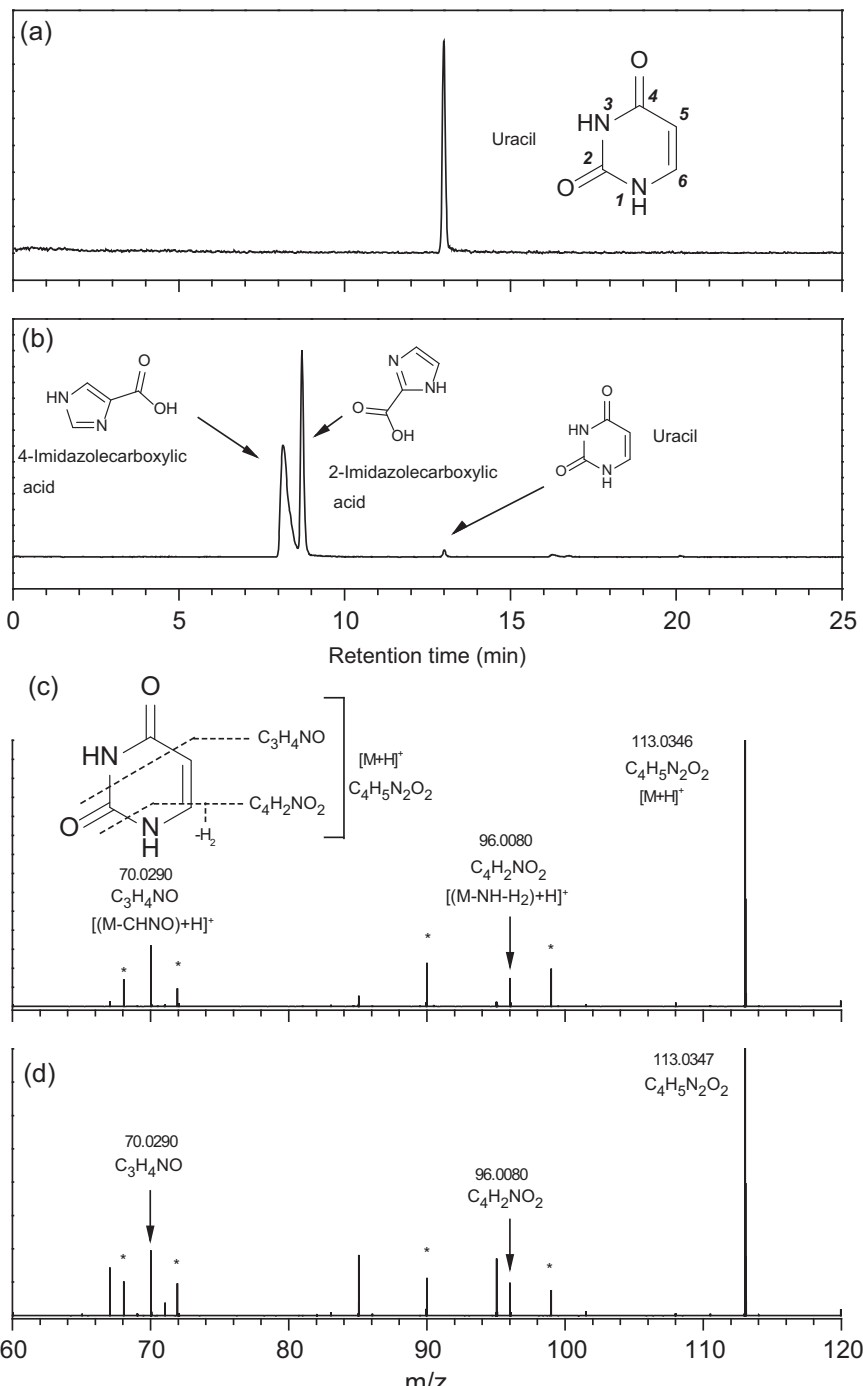

**Fig. 2 Detection of pyrimidine nucleobases.** Mass chromatograms at $m/z = 113.0346$ within a 3 ppm exact mass window at each monoisotopic mass for **a** the uracil (0.46 ng) standard reagent (STD) and **b** Murchison #1 extract measured using a Hypercarb column. In addition to uracil, 4-imidazolecarboxylic acid and 2-imidazolecarboxylic acid were detected as the dominant structural isomers in the Murchison #1 extract. Mass fragmentation patterns of **c** uracil STD and **d** uracil were detected in the Murchison #1 extract measured by MS/MS experiments. Mass peaks with a star (*) represent background signals in the fragmentation spectra present in both the meteorite and reference STD spectra. Bold italic numbers in panel **a** represent the numbering of atoms in the chemical structures of pyrimidine nucleobases.

sample. This would also be the case for nucleobases. Moreover, in general, the higher concentration of dihydrouracil compared to nucleobases cannot be explained by terrestrial biology[35]. Hence, we conclude that the observed chemical diversity of the purine and pyrimidine molecules and their isomers are indigenous and suggests an extraterrestrial origin, consistent with the detection of racemic amino acids and primordial sugars from the same extracts[6,7].

**Purine and pyrimidine molecules and their isomers in the Tagish Lake and Murray extracts.** Various purine and pyrimidine molecules were also identified in the Tagish Lake and Murray extracts (Tables 1 and 2). Guanine was the most abundant in the Murray extract (25 ppb), as is the case for the two Murchison extracts, whereas, for the Tagish Lake extract, adenine was the most abundant (6 ppb) purine molecule and 6-methyluracil (17 ppb) is the most abundant pyrimidine

**Table 2 Variations in the concentration (in ppb) of pyrimidine nucleobases and their related analogs in meteorite extracts and the Murchison soil.**

| | Molecular formula | m/z (M + H⁺) | Murchison | | | | Tagish Lake | Murray | | Murchison soil |
|---|---|---|---|---|---|---|---|---|---|---|
| | | | #1 | #2 | Weighted average | ref. [29] | | This study | ref. [29] | |
| **Cytosine & its isomers** | | | | | | | | | | |
| Cytosine | $C_4H_5N_3O$ | 112.0505 | 4 | 4 | 4 | | 5 | 2 | | 30 |
| Isocytosine | $C_4H_5N_3O$ | 112.0505 | 0.5 | 1 | 1 | | 1 | 0.2 | | 0.2 |
| 1H-Imidazole-4-carboxamide | $C_4H_5N_3O$ | 112.0505 | 5 | 3 | 4 | | 1 | 6 | | n.d. |
| **Uracil & its isomers** | | | | | | | | | | |
| Uracil | $C_4H_4N_2O_2$ | 113.0346 | 15 | 1 | 7 | 30 | n.d.** | n.d. | 19 | <138 |
| 2-Imidazole carboxylic acid | $C_4H_4N_2O_2$ | 113.0346 | 19 | 9 | 13 | | 4 | 14 | | n.d. |
| 4-Imidazole carboxylic acid | $C_4H_4N_2O_2$ | 113.0346 | 90 | 54 | 70 | | 19 | 99 | | <20 |
| **Methyl cytosine** | | | | | | | | | | |
| 5-Methylcytosine | $C_5H_7N_3O$ | 126.0662 | 1 | 1 | 1 | | n.d. | 1 | | n.d. |
| 6-Methylisocytosine | $C_5H_7N_3O$ | 126.0662 | 0.2 | 0.4 | 0.3 | | n.d. | 1 | | n.d. |
| 5-Methylisocytosine | $C_5H_7N_3O$ | 126.0662 | 0.1 | 0.1 | 0.1 | | n.d. | | | |
| **Thymine & its isomers** | | | | | | | | | | |
| Thymine | $C_5H_6N_2O_2$ | 127.0502 | 5 | 1 | 3 | <2 | 5 | n.d. | ≤1 | <19 |
| 6-Methyluracil | $C_5H_6N_2O_2$ | 127.0502 | 4 | 0.5 | 2 | | 17 | 0.3 | | <12 |
| 1-Methyluracil | $C_5H_6N_2O_2$ | 127.0502 | 1 | 5 | 3 | | n.d. | 5 | | |
| 1-Methyl-1H-imidazole-5-carboxylic acid | $C_5H_6N_2O_2$ | 127.0502 | 150 | 75 | 107 | | 9 | 100 | | |
| 2-Methyl-1H-imidazole-4-carboxylic acid | $C_5H_6N_2O_2$ | 127.0502 | 21 | 11 | 15 | | 9 | 23 | | |
| 4-Methyl-1H-imidazole-5-carboxylic acid | $C_5H_6N_2O_2$ | 127.0502 | 26 | 22 | 24 | | n.d. | 30 | | |
| **Others** | | | | | | | | | | |
| 4-Hydroxypyrimidine | $C_4H_4N_2O$ | 97.0396 | 6 | 2 | 4 | | 10 | 3 | | 23 |
| 2,4-Diaminopyrimidine | $C_4H_6N_4$ | 111.0665 | 1 | n.d. | 0.2 | | n.d. | 0.2 | | n.d. |
| 5-amino-4-imidazolecarboxamide | $C_4H_6N_4O$ | 127.0614* | 0.02 | *** | | | | | | n.d. |
| Dihydrouracil | $C_4H_6N_2O_2$ | 115.0502 | 51 | n.d. | 25 | | n.d. | n.d. | | n.d. |
| Nicotinamide | $C_6H_6N_2O$ | 123.0553 | 10 | 20 | 16 | | 5 | 65 | | |
| Isonicotinamide | $C_6H_6N_2O$ | 123.0553 | 3 | 3 | 3 | | 4 | 6 | | |
| Picolinamide | $C_6H_6N_2O$ | 123.0553 | 1 | 2 | 1 | | 3 | 10 | | |
| Nicotinic acid | $C_6H_5NO_2$ | 124.0393 | 91 | 264 | 190 | | 108 | 626 | | |
| Isonicotinic acid | $C_6H_5NO_2$ | 124.0393 | 53 | 216 | 147 | | 118 | 307 | | |
| | | Total pyrimidines | 37 | 15 | 24 | | 38 | 13 | | |
| | | Pyrimidines/Purines | 0.2 | 1.4 | 0.3 | | 4.3 | 0.4 | | |

*This is the m/z of the negative ion (M-H)⁻ of 5-amino-4-imidazolecarboxamide.
**n.d. not detected.
***Blank means "not searched for" in the present study, and "not reported" in previous studies.

molecule. The Tagish Lake extract generally shows less diversity of purine and pyrimidine molecules compared with the Murchison and Murray extracts, and their absolute concentrations are generally lower than those of the other meteorite extracts. The observed trend in molecular abundance and diversity is consistent with that observed for amino acids in these three meteorites[28], suggesting that the Tagish Lake and the other two meteorites may have experienced distinct processes on their parent bodies. The total concentrations of detected purine molecules in the Tagish Lake and Murray extracts were 9 and 33 ppb, respectively, and those of pyrimidine molecules were 37 and 13 ppb, respectively. Note that there is a prominent difference in the (total pyrimidine molecules)/(total purine molecules) ratio between the Tagish Lake and the other two meteorites: for the former, pyrimidine molecules dominate over purine molecules, whereas the opposite is true for the latter. This observed difference may reflect different formation pathways, as discussed later.

**Important nitrogen heterocyclic molecules in the meteorite extracts.** Imidazole is an interesting molecule in terms of a primitive catalyst for molecular syntheses[36], which might be a potential precursor of purine formation[37]. Due to its potential importance, it has been a target of multiple astrochemical[25,37] and astronomical studies[38]. Here, we identified not only imidazole itself but also its related analogs, some of which are the

structural isomers of pyrimidine nucleobases such as cytosine, uracil, and thymine at typically higher concentrations than the corresponding pyrimidine bases (Fig. 2 and Supplementary Fig. 3). For example, 4-imidazolecarboxylic acid and 1-methyl-1H-imidazole-5-carboxylic acid, which are structural isomers of uracil and thymine, respectively, are 1–2 orders of magnitude more abundant than the corresponding pyrimidine nucleobases (Table 2). This strongly suggests the presence of preferential routes for the formation of imidazole-containing molecules over pyrimidine-containing molecules in extraterrestrial environments. The reactivity of imidazole and the contribution of electron-conjugated systems are assumed to induce progressive cyclization and variation in purine derivatives. Indeed, imidazole and its alkylated homologs are much more abundant than pyrimidine nucleobases (Table 3), supporting our hypothesis. We estimated that the progression of the N-heterocyclic reaction would correlate with the thermal history of the parent body.

We also confirmed the presence of alkylated homologs of imidazole using the same procedure described above (Supplementary Fig. 6). Larger alkylated imidazole homologs were observed in the mass chromatograms using a retention index of systematic methylene (-CH₂-) elongation (Supplementary Fig. 7). These results are qualitatively consistent with the tentative detection of alkylated imidazole homologs (i.e., imidazole with <C₁₅) in the Murchison meteorite[9]. Given that the observed peaks

**Table 3 Variations in the concentrations (in ppb) of imidazole and its alkylated analogs.**

| Name | Molecular formula | $m/z$ (M + H$^+$) | Murchison | | | Tagish Lake | Murray |
|---|---|---|---|---|---|---|---|
| | | | #1 | #2 | Weighted average | | |
| Imidazole | $C_3H_4N_2$ | 69.0447 | 96 | 32 | 59 | 82 | 51 |
| 4(5)-Ethylimidazole | $C_5H_8N_2$ | 97.0760 | 21 | 66 | 47 | 2 | 15 |
| 1,2-Dimethylimidazole | $C_5H_8N_2$ | 97.0760 | 57 | 53 | 55 | 4 | 4 |
| 2-Ethylimidazole | $C_5H_8N_2$ | 97.0760 | 12 | 12 | 12 | 0.4 | 8 |
| 2-Isopropylimidazole | $C_6H_{10}N_2$ | 111.0917 | 4 | 9 | 7 | n.d. | 6 |
| 2-Ethyl-4-methylimidazole | $C_6H_{10}N_2$ | 111.0917 | 85 | 95 | 91 | 2 | 18 |
| 2-Propylimidazole | $C_6H_{10}N_2$ | 111.0917 | 3 | 17 | 11 | 1 | 5 |
| | | Total | 278 | 284 | 281 | 91 | 106 |

on each mass chromatogram are solely derived from alkylated imidazole homologs with identical ionization efficiencies, the total concentrations of alkylated imidazole homologs could be as high as 41,300, 500, and 21,200 ppb in the Murchison, Tagish Lake, and Murray extracts, respectively. This implies that alkylated imidazole homologs are one of the most abundant classes of solvent-extractable organic compounds in these three carbonaceous meteorites. Naraoka et al.[9] proposed a possible pathway for the formation of alkylimidazoles via the Radziszewski reaction, which uses aldehydes and ammonia as the initial reactants. Since aldehydes and ammonia might be formed by the hydrothermal decomposition of HMT in meteorite parent bodies[26], the formation of alkylimidazoles could be enhanced under the same environmental conditions.

Figure 3 shows the variations in the relative abundances of alkylated imidazole homologs in the meteorite extracts with respect to the number of carbon atoms in the alkyl groups under the above assumptions. The variation in the Tagish Lake extract is clearly distinct from that in the other two meteorite extracts, suggesting a different parent body process. In contrast to the predominance of imidazole and its homologs, pyrazole (Supplementary Fig. 1), the structural isomer of imidazole, was not detected in any of the meteorite extracts. Laboratory experiments by Vinogradff et al.[39] on the hydrothermal degradation of HMT in the presence of phyllosilicates (~150 °C for up to 31 days) provided various alkylated pyrazoles. We observed several unidentified peaks for the structural isomers of the alkylated imidazole and/or pyrazole (Supplementary Fig. 7), implying that alkylated pyrazole homologs were also present in the meteorite extracts. We also tentatively observed N-containing compounds with a linear structure (see Supplementary Text and Supplementary Fig. 8), where the hypothesized concentration trend between meteorites follows the same trend as that of alkylimidazoles (Supplementary Fig. 9), suggesting a distinct molecular distribution between the Tagish Lake meteorite and the other two meteorites.

Supplementary Fig. 10 shows mass chromatograms at $m/z$ 123.0553 and 124.0393, which corresponds to those of the protonated ions of nicotinamide and nicotinic acid, respectively, in the Murchison #1 extract. Based on their chromatographic retention time and accurate mass measurements of their parent masses, we identified two molecules in each meteorite extract. Both molecules have a pyridine structure with a carboxyl or amide group (Supplementary Fig. 10) and are generally classified as vitamin B$_3$. Nicotinic acid has been identified in various carbonaceous meteorites with concentrations on the order of hundreds of ppb[40], which is consistent with the results of the present study (Table 2). On the other hand, nicotinamide was not found in the hot water extract (at 100 °C for 24 h), acid-hydrolyzed hot water extract, or formic acid extract[40]. Under these severe extraction conditions, nicotinamide could have been

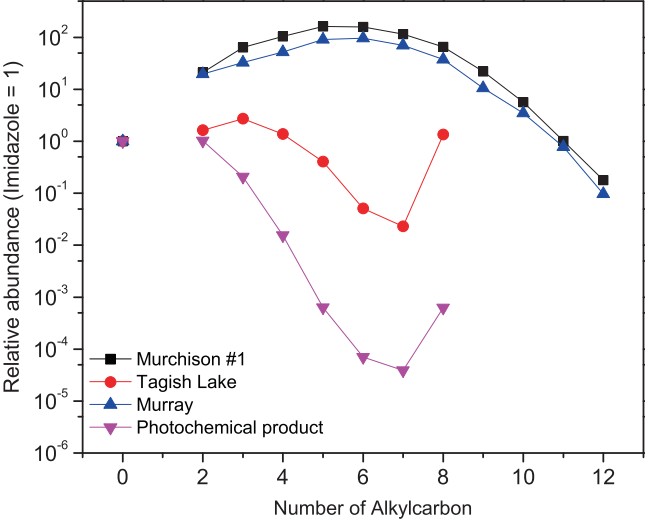

**Fig. 3 Possible relative abundances of alkylated imidazole analogs.** Variations in the hypothesized relative abundances of imidazole and its alkylated homologs in the Murchison #1, Tagish Lake, and Murray extracts, with relevance to the number of alkylcarbons as [(-CH$_2$-)$n$]. The relative abundances of these molecules in the organic residues formed by photochemical reactions of interstellar ice analogs are shown for comparison.

hydrolyzed to yield nicotinic acid. Because the extraction method we used in our study was much gentler, as explained in the methods section, we believe that it may be more suitable for extracting such an acid-labile molecule. Supplementary Table 2 shows a relative abundance of nicotinic acid and nicotinamide (acid/amide) in each meteorite extract, as well as that value in the laboratory-made photochemical products prepared in our previous study[25]. Nicotinic acid was more abundant than nicotinamide in all meteorite extracts, but the degree varied between meteorites. For both Murchison and Murray extracts, the acid/amide ratio was ~10, whereas it was ~20 for the Tagish Lake extract. Since Tagish Lake meteorites are considered to have experienced a range of aqueous alteration conditions on the parent body[41], the high acid/amide ratio may be indicative of the parent body processes. In fact, the acid/amide ratio for the photochemical products, which did not undergo any heating processes[25], showed an acid/amide value of 0.8, supporting the above assumption. Further experiments, such as hydrolysis of nicotinamide, may be helpful to better understand the potential of the nicotinic acid/nicotinamide ratio for estimating the degree of aqueous alteration in meteorite parent bodies. In addition to nicotinic acid and nicotinamide, their structural isomers, such as isonicotinic acid and isonicotinamide, respectively, were identified in the same mass chromatograms (Supplementary Fig. 10),

which indicates a wide diversity of organic molecules in the meteorite extracts, as described in the next section.

**Wider diversity of pyrimidine molecules compared to previous studies.** The detection of diverse suites of pyrimidine nucleobases and their analogs is distinct from previous reports on the detection of nucleobases in carbonaceous meteorites (ref. [11] and references therein). One possible reason for this might be the sample heterogeneity between the Murchison #1 and Murchison #2 meteorites (Table [2]). However, the observed heterogeneity mainly arises from the concentration of the detected pyrimidine molecules rather than molecular diversity. We are surely certain that analytical and instrumental developments applied in the present study have a significant advantage for detecting ppb to ppt (= nanogram to picogram per gram-meteorite) orders of pyrimidine molecules in meteorite extracts. For example, ion-exchange chromatography as a desalting process improved after Takano et al.[42] should have resulted in the higher detectability of these molecules by removing coexisting molecules that potentially interfere with the detection of pyrimidine molecules[25]. In addition, the HPLC conditions (e.g., the use of formic acid in the HPLC solvent as a protonated reagent), which are distinct from those applied in previous studies (for example, ref. [43]) may have enhanced the detectability of these molecules. In fact, under the present analytical conditions, even for sub-nanogram-order nucleobases, their peaks are outstanding (Figs. [1] and [2]), which indicates a much higher detectability of nucleobases.

**Possible contribution from photochemical reactions in the ISM to the distribution of N-heterocyclic molecules in meteorites.** The above discussion provides robust evidence that the nucleobases and their analogs detected in meteorite extracts are of exogenous origin with a unique molecular history. We previously demonstrated that pyrimidine and purine nucleobases can be formed by photochemical reactions of ice mixtures containing water, carbon monoxide, methanol, and ammonia under interstellar medium (ISM) conditions[25]. Although the detailed formation pathways by photochemical reactions are currently not well characterized, formamide ($NH_2CHO$) is also a possible source of nucleobase formation in extraterrestrial environments[22,25]. It is noteworthy that photochemical reactions of interstellar ice analogs yield pyrimidine nucleobases one order of magnitude more abundant than purine nucleobases, but even pyrimidine nucleobases are minor compared to their structural isomers in the product[25]. These molecular properties of photochemical products (Supplementary Table 3) closely resemble those of the Tagish Lake extract, as shown by the high relative abundance of pyrimidine molecules compared to purine ones (Supplementary Fig. 11). In addition, the possible distributions of alkylated imidazole homologs in the Tagish Lake extract were similar to those of photochemical products prepared in previous studies (Fig. [3])[25]. These similarities imply that photochemical reactions in the ISM may partly contribute to the presence of nucleobases and related molecules in the Tagish Lake meteorite. An interstellar origin of meteoritic organic molecules was also inferred for HMT detected in the Murchison, Tagish Lake, and Murray meteorites[26].

It is highly probable that, in addition to the pathways proposed above, there should be other routes to the formation of purine and pyrimidine nucleobases present in carbonaceous meteorites (see Supplementary Text). In addition, nucleobases with amino groups may easily decompose by hydrothermal activities on the meteorite parent bodies, and some molecules yield their oxygenated counterparts (e.g., from cytosine to uracil)[33]. A full understanding of the formation pathways of purine and pyrimidine nucleobases in carbonaceous meteorites is beyond the scope of the present study. Nevertheless, we

believe that the detection of various molecules capable of acting as intermediates or precursors for nucleobase synthesis provides a significant advance in our understanding of prebiotic chemistry in meteorites.

**Future perspectives and possible relevance to the origin of life on the early Earth.** We reported the detection of more than 30 nitrogen heterocyclic molecules, including several DNA/RNA nucleobases in the meteorite extracts; however, a number of related molecules remain unidentified in the meteorite extracts (e.g. Supplementary Fig. 3c). Bulk analysis of high-resolution mass spectra of meteorite extracts by computational operations such as ATTRIBUTOR software[10] can help identify families of homologous compounds present in meteorite extracts and thus can be used on our target molecules. Moreover, purely computational approaches, such as ab initio molecular dynamics simulations, to search for chemical reaction pathways in a complex system, have been developed in the last decade[44,45]. Looking ahead to the next decade in combination with laboratory-based analysis and model simulation studies, we are now approaching a more complete understanding of the molecular distributions of meteoritic organics, including nucleobases and their formation pathways. In this context, it is highly anticipated that various kinds of nucleobases and their analogs can be detected from the samples returned from the carbonaceous asteroids Ryugu[1,46,47] and Bennu[2].

The present study detected diverse suites of both purine and pyrimidine nucleobases, including canonical base pairs (e.g., adenine-uracil, guanine-cytosine, adenine-thymine) and some non-canonical ones (e.g., isoguanine-isocytosine and xanthine-2,4-diaminopyrimidine)[48] in carbonaceous meteorites (Tables [1] and [2]). Given that extraterrestrial materials, including meteorites, were provided to the Hadean Earth at a flux much higher than that in the present day[12], a large number of these canonical and non-canonical base pairs may have also been delivered to the Earth at that time. Although the formation of these nucleobases from concentrated source materials such as HCN, formamide, urea, and from highly reduced atmospheres have been reported in previous studies[21,24], the accumulation of these scarce molecules has substantial geochemical challenges on Hadean Earth with an atmosphere possibly dominated by $CO_2$ and $N_2$ ref. [49]. Hence, we expect that the exogenous base pairs contributed to the emergence of genetic properties for the earliest life on Earth.

## Methods

**Meteorites.** Two specimens of the Murchison meteorite (CM2) were used. One specimen was stored at the University of Chicago for many years in a desiccator until it was removed in August 2015 ref. [34]. This 10 g specimen was crushed and homogenized at the NASA Goddard Space Flight Center, and a 2 g portion of the powder was used for extraction of the nucleobases. Sugars such as ribose and arabinose[6] and hexamethylenetetramine (HMT)[26] have been recently identified in aqueous extracts. The second Murchison specimen, as well as specimens of the Murray (CM2) and Tagish Lake (C2 ungrouped) meteorites, were certified by and purchased from a meteorite trading company and used for the analysis of nucleobases. To the best of our knowledge, unlike the Murchison and Murray meteorites, there are no reports regarding the detection of nucleobases or other nitrogen heterocyclic molecules from the Tagish Lake meteorite. The clean-up procedure for these meteorites was described by Oba et al.[26]. The Murchison specimens from the University of Chicago and those from the meteorite trading company are hereinafter denoted as Murchison #1 and Murchison #2, respectively.

**Extraction of nucleobases and nitrogen heterocyclic molecules from meteorites.** We searched for various nucleobases and their structural isomers, as well as other nitrogen heterocyclic molecules (Supplementary Fig. 1). These molecules were extracted from the cation-desalting fraction of the Murchison #1 extract as described by Furukawa et al.[6]. Finely powdered samples of the Murray, Tagish Lake, and Murchison #2 meteorites were subjected to water extraction (2.7 g for Murchison #2, 2 g for Murray, and 0.5 g for Tagish Lake) using ultra-sonication

(10 min with crushed ice in the sonic bath) with two bed-volume of ultrapure water at room temperature (qTOF grade, Fujifilm Wako Co. Ltd). After solid/liquid separation by centrifugation for 10 min (1610×g), the supernatant was recovered. This procedure was repeated three times and the extracted liquid was combined. The extract was frozen and dried under reduced pressure at ambient temperature. Inorganic salts and interfering organic matrix from the extracts were removed by isolating the nucleobase-containing fraction using cation exchange chromatography (AG50W-X8 resin, Bio-Rad Laboratories)[42]. The final eluate containing nucleobases and N-heterocyclic molecules was dried, dissolved in 1 mL of ultrapure $H_2O$, and filtered using a 0.20 μm PTFE cartridge filter. The recovery of nucleobases and other N-heterocyclic derivatives using this procedure was previously confirmed using standards established by Oba et al. (>95.0 ± 3.5%, $n = 3$)[25]. All glassware was cleaned by heating in air at 450 °C for 3 h prior to being used for sample processing and molecular analysis.

**Synthesis of photochemical products**. We analyzed the organic residues produced by the photolysis of ice mixtures containing $H_2O$, $CO$, $NH_3$, and $CH_3OH$ under astrophysically relevant conditions at 10 K. Details of the preparation scheme have been reported in a previous study[25]. To summarize, the ice mixture was photolyzed on an aluminum substrate at 10 K in an experimental setup (SAMRAI) equipped with a stainless-steel chamber, two deuterium lamps, a quadrupole mass spectrometer, a turbo molecular pump, and a helium cryostat. After photolysis, the sample was warmed to room temperature and the formed organic residues were extracted with ~0.5 mL of a mixture of $H_2O/CH_3OH$ (1/1, v/v). The extracted samples were treated using the same procedure as described above.

**Chromatographic separation and time-based identification of nucleobases by HPLC**. Some heterocyclic compounds have the same chemical composition and exact mass number (e.g., cytosine and isocytosine; guanine and isoguanine); therefore, chromatographic separation is an important process for the optimization of online mass spectrometry. The purified fraction of each sample was introduced into an HPLC/ESI-HRMS instrument with a mass resolution of 140,000 at a mass-to-charge ratio ($m/z$) of 200. The online system comprised an UltiMate 3000 and Q-Exactive Plus (Thermo Fischer Scientific) equipped with a reversed-phase separation column (InertSustain PFP, 2.1 × 250 mm, particle size 3 μm, GL Science or HyperCarb™, 2.1 × 150 mm, particle size 5 μm, Thermo Fischer Scientific) at 40 °C and operated in positive ion mode. Validations of analytical methods were also performed using a 1290 Infinity II coupled with a 6230 time-of-flight mass spectrometer (Agilent Technologies). The eluent program for the HPLC setup with the PFP column was as follows: at $t = 0$ to 5 min, solvent A (water), solvent B (acetonitrile + 0.1% formic acid) = 90:10, followed by a linear gradient of A:B = 50:50 at 20 min and maintained at this ratio for 25 min. The flow rate was 0.1 mL/min. The eluent program for the HyperCarb™ column was as follows: solvent A (water + 0.1% formic acid) and solvent B (acetonitrile + 0.1% formic acid) = 100:0 at $t = 0$ min, followed by a linear gradient of A:B = 75:25 at 20 min and maintained at this ratio for 15 min.

We used an improved gradient program to maintain ionization efficiency because the addition of 0.1% formic acid (as a proton donor) to solvents A and B created an inverse elution and ensured the stability of the electron spray ionization efficiency. The flow rate was constantly 0.2 mL/min. The PFP column was especially suitable for the chromatographic separation of purine nucleobases, but was inappropriate for the analysis of pyrimidine nucleobases because co-eluting structural isomers prevented some target molecule identification and quantification. On the other hand, the use of HyperCarb™ has a strong advantage for the separation of pyrimidine nucleobases because of the appropriate affinities between nitrogen-containing molecules and the stationary phase. Molecules that were not detected in positive ion mode, such as 5-amino-4-imidazolecarboxamide (Supplementary Fig. 12) and 5-aminoimidazole-4-carbonitrile (Supplementary Fig. 13), were analyzed in negative ion mode using the same instrument but with a HILICpak VG-50 column (2.1 × 150 mm, particle size 5 μm, Shodex) at 40 °C. The elution program was as follows: solvent A ($H_2O$ + 0.5% ammonia) and solvent B (acetonitrile) = 20:80 at $t = 0$–5 min, followed by a linear gradient of A:B = 90:10 at 35 min and maintained at this ratio for 10 min. The flow rate was 0.2 mL/min. High-purity grade water and acetonitrile (LC/MS grade from Fujifilm Wako Chemical, Ltd.) were used to minimize analytical noise and background signals in the LC and Orbitrap MS.

**Exact mass identification of nucleobases by HRMS**. The mass spectra were recorded in positive or negative ESI mode with an $m/z$ range of 50–400 and a spray voltage of 3.5 kV. We confirmed the MS response linearity over a dynamic range of substrate concentrations on the order of ppt, ppb, and ppm scales[25,26]. The exact mass was obtained with a general mass accuracy of 3 ppm, defined as [(measured $m/z$) – (calculated $m/z$)]/(calculated $m/z$) × $10^6$ (ppm). The separated compound solution was heated at 300 °C for desolvation using a HESI-II (Thermo Fischer Scientific) or Agilent Jet Stream (Agilent Technologies). The capillary temperature of the ion transfer was 300 °C. The injected samples were vaporized at 300 °C.

MS/MS experiments were also performed using a hybrid quadrupole-Orbitrap mass spectrometer (Q-Exactive Plus, Thermo Fisher Scientific) with HPLC and ionization conditions identical to those used for the full-scan analyses. The

extracted positive ion $m/z$ (for example, for thymine, 127.05 ± 0.2) was reacted with high-energy collision $N_2$ gas to produce fragmented ions, and the mass range of $m/z$ 50–160 was monitored using an Orbitrap MS with a mass resolution of ~140,000. MS/MS measurements were not performed for some nucleobases because of their low concentrations in the extracts.

Standard reagents for the target molecules (Supplementary Fig. 1) were purchased from Combi-Blocks Inc., Toronto Research Chemicals, Fujifilm Wako Chemical, Ltd., Tokyo Chemical Industry, Ltd., Sigma-Aldrich Ltd., and BLD Pharmatech, Ltd., and used without further purification.

**Procedural blank test**. The analytical blank test was performed using 2 g of combusted quartz subjected to the same extraction procedure described above to evaluate possible contamination by nucleobases during sample preparation. In addition, we analyzed a soil sample (102 mg) collected with a clean metal scoop from a depth of 20–30 cm from the Murchison meteorite-strewn field in 1999 to assess the potential terrestrial contamination of the Murchison meteorite from the impact site. Nucleobases and their related molecules extracted from the soil were analyzed using the same analytical setup with a PFP separation column. Further details are provided in the supplementary information of Furukawa et al.[6] and Oba et al.[26].

## Data availability

The data that support the findings of this study are available from the corresponding author (Y.O.) upon reasonable request.

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

## Acknowledgements

We thank Dr. Shogo Tachibana (Univ. Tokyo, JAXA) for the discussion of meteoritic organic matter within the thermal history. We thank Dr. Robert Minard and Dr. Clifford N. Matthews' research group at the University of Chicago for providing the Murchison meteorite and Prof. Reid R. Keays (Univ. Melbourne) for collecting and providing the Murchison soil samples. We also thank Mr. Tomoya Ishida (Kyushu Univ.) for the discussion of the wet-chemical procedure. This work was partly supported by JSPS KAKENHI Grant Numbers JP21H04501 and JP21H05414 (to Y.O.), JP20H02019 and 21KK0062 (to Y.T.), 21J00504 (to T.K), JP20H00202 and JP20H05846 (to H.N.), NASA Astrobiology Institute through award 13-13NAI7-0032 to the Goddard Center for Astrobiology (GCA), NASA's Planetary Science Division Internal Scientist Funding Program through the Fundamental Laboratory Research (FLaRe) work package at NASA Goddard Space Flight Center, and a grant from the Simons Foundation (SCOL award 302497 to J.P.D.). This study was conducted in accordance with the Joint Research Promotion Project at the Institute of Low-Temperature Science, Hokkaido University (21G008 to Y.O., Y.T., and H.N.).

## Author contributions

Y.O. and Y.T. designed this project, outlined the development with H.N. For sample preparation, D.P.G. and J.P.D. performed the pretreatment and quality assessment of the Murchison sample from Univ. Chicago. Y.T. and Y. F. extracted indigenous nitrogen compounds from meteorites and purified them before LC/ESI-HRMS analysis. Y.O., Y.T., Y.F., and D.P.G. conducted an assessment using a reference soil sample from the Murchison meteorite fall locality in Murchison, Australia. Y.O., T.K., and H.N. analyzed the samples with authentic standards and reference materials. Y.O. prepared the manuscript. All authors commented on the manuscript.

## Competing interests

The authors declare no competing interests.
