## [Peer Review File · Nature Communications]

Identifying the wide diversity of extraterrestrial purine and pyrimidine nucleobases in carbonaceous meteoritesREVIEWER COMMENTS

Reviewer #1 (Remarks to the Author):

Dear Dr. Oba and co-authors,

This manuscript presents the results obtained from the analysis of N-containing heterocyclic molecules extracted from Murchison, Murray and Tagish Lake carbonaceous meteorites using a high-performance liquid chromatography coupled with electrospray ionization high-resolution mass spectrometry (HPLC/ESI-HRMS). In addition to the compounds previously detected, such as guanine and adenine, various pyrimidine nucleobases such as cytosine, uracil, and thymine, and their structural isomers, which most of which had not been previously detected in meteorites, were detected. The concentration level ranges up to parts per trillion. The molecular distribution of pyrimidines in meteorites were similar to those produced from the synthesis in simulated interstellar ice analogues, it is discussed that some of these derivatives could have been generated by photochemical reactions in the interstellar medium. The authors' study is very interesting and important for enhancing our knowledge of the diverse synthetic pathways of nucleobases in the early Solar System. I would recommend the manuscript for acceptance after the minor revision. My questions and comments are summarized below. Hope that some of these are helpful for improving the manuscript.

- Page 6, Lines 101-102

Would you discuss why Guanine is not the most abundant in Tagish Lake meteorite? Is it explainable by Fig. 3 as well?

- Page 10, Lines 186-188

"The absence of pyrimidine in meteorite extracts suggests that the detected pyrimidine nucleobases and their analogs are not formed by the addition of side chains such as methyl (-CH₃) and amino (-NH₂) groups."

The conclusion sounds a bit strong. Given that the concentrations of the pyrimidine nucleobases are so low, is there possibility that pyrimidine is under detection limit?

- Page 12, Lines 230-232

It is suggested that the Tagish Lake and the other two meteorites (Murchison and Murray) may have experienced distinct processes on their parent bodies, since the total concentrations of detected purine molecules in the Tagish Lake and Murray extracts were 9 and 33 ppb, respectively, and those of pyrimidine molecules were 37 and 13 ppb, respectively.

However, the total concentrations of detected purine molecules in the Tagish Lake and (either of) Murchison extracts were mostly consistent, 9 and 11 ppb, respectively, and those of pyrimidine molecules were also consistent, 38 and 37 ppb, respectively. This indicates that differences of the total concentrations of purines and pyrimidines are within the ranges of heterogeneity which the authors discuss.

The authors also mention that the pyrimidines/purines ratios are distinct between Tagish Lake and the other two meteorites. However, the ratio of Murchison #2 is seven times higher than that of Murchison #1, while the ratio of Tagish Lake is only three times higher than that of Murchison #2. The difference appears to be within the ranges of heterogeneity. Thus, it seems that the total concentrations and their ratios cannot necessarily explain the distinct processes on the meteorite parent bodies.

Rather, Supplementary Figs 8 and 10 are more convincing to explain the difference between Tagish Lake and the other meteorites. I would suggest that Supplementary Fig. 8 should be shown in the main manuscript.

- Supplementary Fig 8

In the relative abundances of imidazole alkyl analogues of Tagish Lake meteorite and the photochemical product, why C8-alkyl imidazole is relatively high abundant?

- Page 13, Fig. 3

Isn't there a possibility that Xanthine is produced from Guanine by deamination during meteorite parent body alteration?

The authors support the synthetic pathways in Fig. 3 by mentioning that the presence of 5-aminoimidazole-4-carbonitrile was not detected but highly expected. On the other hand, they exclude the synthetic pathway involving the compound (pyrimidine) which they did not detect. This seems to lack consistency in data interpretation and puts their own spin on discussion.

- Page 18, Lines 331-339

It is better to bring this part to the end of the manuscript. I am wondering how effectively such low concentration level (ppt) of nucleobases in meteorites contributed to life's building blocks, in comparison to those produced by the other pathways under different environments, although I am not going to deny your hypothesis...

- Page 21, Line 399 Meteorites

If one can evaluate the extent of alteration for the Tagish Lake meteorite used in this study (by comparison with the different lithologies of Tagish Lake meteorite used in Herd et al (2011)), it would be helpful to discuss the relationship between molecular distributions of nucleobases and aqueous alteration.

Minor comments:

- Page 3, Lines 58-60 "A substantial amount of exogenous meteoritic organics..."
References are needed.

- Page 9, Lines 162-178

Description of future study is lengthy. Would you please shorten this paragraph.

- Page 9, Line 175

"heavy isotope enrichment in extraterrestrial nucleobases may not be indicative of their extraterrestrial origin."

-> "heavy C isotope enrichment in extraterrestrial nucleobases may not be indicative of their extraterrestrial origin." (because the previous study examined only carbon isotope fractionation)

- Table 1

Please annotate "uc" and "mtc".

Reviewer #2 (Remarks to the Author):

The authors describes the identification of pyrimidine nucleobases in organic (formic acid) extracts from the Murchison, Murray, and Tagish Lake meteorites, using high-performance liquid chromatography coupled with electrospray ionization high-resolution mass spectrometry (HPLC/ESI-HRMS). Sample heterogeneity in the organic composition of two Murchison's samples has been highlighted. The novelty of the study is the "first-time" identification of two pyrimidine nucleobases, thymine and cytosine, besides to previously detected uracil, thus completing the panel of canonical pyrimidine nucleobases required for the emergence of primordial DNA and RNA. Others pyrimidine heterocycles and purine nucleobases were detected and quantified, including (for the pyrimidine type) isocytosine, 5-methylcytosine, 6-methylisocytosine and 5-methylisocytosine. In the last part of the manuscript, the detection of pyrimidine nucleobases is associated to photochemical models describing possible synthetic pathway for these compounds.

It is my opinion that the manuscript contains some novelty elements to be published as a communication in Nature, although it requires an extensive reworking and partly revision of the text before the publication (major revision).

First, the overall description of the study is too long and should be reduced significantly. I find the

description of the analytical composition of the extracts (first section of the manuscript) the clearest part of the study.

Some corrections are required.

They include the revision of the chemical language (and corresponding chemical structures). For example, the definition of pyrimidine nucleobases as “single 6-membered nitrogen (N) heterocyclic ring” is too general. In canonical DNA/RNA pyrimidine nucleobases the nitrogen atoms are always in 1,3 positions and the heterocyclic nucleus of uracil and thymine should be more appropriately reported as 1(H),3(H)-pyrimidin-2,4-dione for etc. The same consideration is valid for purine nucleobases.

In figures (text and supplementary information) nucleobases are some-time represented as the less thermodynamically stable tautomeric form instead of the more stable one [e.g. N9(H) for adenine etc].

The numbering of atoms in the chemical structures of nucleobases may be useful for a better reading of the text.

Figure 3 is unnecessary.

In the second part, I find the discussion on the possible mechanism of formation of pyrimidine nucleobases very speculative and incomplete. I mean, the sentence “photochemical reactions in the ISM may partly contribute to the presence of nucleobases and related molecules in the Tagish Lake meteorite (pag 19, 357)”, as well as the sentence “The distinct variation pattern of alkylated amines from that of amino acids and imidazoles might imply their different formation processes during the formation of the early solar system (line 33, supplementary information), imply that nucleobases are simply stored in meteorites and does not take into due account recent studies showing that meteorites are catalytically active under different energy conditions and can both favor the post-ante formation of nucleobases (and other compounds) and lead to their partial or complete degradation (see for example: Chem. Commun., 2019,55, 10563-10566; PNAS J2015, 112, 7109-7110; Chem. Eur. J. 2013, 19, 16916-16922 etc).

In a similar way, the sentence “the validity of the route for the formation of cytosine and uracil starting from cyanoacetylene (HCCCN) (page 20, line 373)” and successive considerations does not take into due account the existence of prebiotic synthetic routes simpler than the use of a complex (astrochemically speaking) three carbon atom precursor such as HCCCN (see: Chem. Eur. J. A 2020, 26, 12075-12080; Chem. Soc. Rev., 2012, 41, 5526–5565 etc).

The next sentence “It is highly probable that, in addition to the pathways proposed above, there should be other routes to the formation of purine and pyrimidine nucleobases present in carbonaceous meteorites” it is inconclusive and does not compensate for the lack of information in the previous section.

I strongly suggest to revise and made more concise the paragraph “Possible formation pathways of nucleobases and their analogs detected in meteorites” limiting it to the essential content and references, reporting on all possible scenarios. Otherwise the paragraph can be completely removed.

In addition, supplementary information should be improved by adding a novel table reporting (for all identified compounds) the intensity and m/z value of the main fragments, highlighting (where necessary) the difference between isomeric structures.

Reviewer #3 (Remarks to the Author):

This is a very interesting paper that reports work investigating the nucleobase contents of carbonaceous chondrites. These compounds are key prebiotic components that could have facilitated the origin of life. The manuscript is well written and gives a good account of previous work. The authors recognised new compounds and identify heterogeneity within samples of the same meteorite.

In Line 177, I wondered if the Callaghan et al. experiments did lead to the decomposition of the compounds the authors mention or whether the current protocol was just conservative, which would be fine but could be unpacked briefly in this paper to justify the different procedure.

The presence of cytosine (Line 156) is interesting and the authors spend some time discussing its possible indigeneity. In their statements I think they mis-cite reference "37" as "38" (Line 175). In this part of the manuscript they argue that heavy isotope "enrichment in extraterrestrial nucleobases may not be indicative of their extraterrestrial origin". This is hard to understand as written. i) It sounds internally contradictory and ii) Nucleobases on Earth are derived from our highly active biosphere and biological nucleobases do not display the enrichment in heavy isotopes apparent in meteoritic compounds. This part of the paper should be made more clear before publication or it runs the risk of confusing the community.

Overall, interesting work with implications for parent body processes and origin of life scenarios. It should generate quite a lot of interest from the readership of Nature Communications.

Replies to the comments by Reviewers

We appreciate the constructive review comments from three Reviewers on our manuscript (NCOMMS-21-41665) entitled “Identifying the wide diversity of extraterrestrial purine and pyrimidine nucleobases in carbonaceous meteorites”. We carefully read the whole comments and modified the original version of the manuscript based on their helpful comments. The changes we made based on the Reviewer’s comments are noted in **red font** in the revised manuscript/supporting information. Our replies to each comment (Times New Roman) are denoted below following the reviewer’s comments (Arial). Note that in our replies below, we denoted Line numbers in the revised manuscript otherwise noted. In supplementary figure 2, we added mass chromatograms of both 2,6-diaminopurine and purine.

Reviewer #1 (Remarks to the Author):

Dear Dr. Oba and co-authors,

This manuscript presents the results obtained from the analysis of N-containing heterocyclic molecules extracted from Murchison, Murray and Tagish Lake carbonaceous meteorites using a high-performance liquid chromatography coupled with electrospray ionization high-resolution mass spectrometry (HPLC/ESI-HRMS). In addition to the compounds previously detected, such as guanine and adenine, various pyrimidine nucleobases such as cytosine, uracil, and thymine, and their structural isomers, which most of which had not been previously detected in meteorites, were detected. The concentration level ranges up to parts per trillion. The molecular distribution of pyrimidines in meteorites were similar to those produced from the synthesis in simulated interstellar ice analogues, it is discussed that some of these derivatives could have been generated by photochemical reactions in the interstellar medium. The authors’ study is very interesting and important for enhancing our knowledge of the diverse synthetic pathways of nucleobases in the early Solar System. I would recommend the manuscript for acceptance after the minor revision. My questions and comments are summarized below. Hope that some of these are helpful for improving the manuscript.

[Reply] Thank you very much for your very positive comments. We carefully read the comments and modified the manuscript as shown below.

- Page 6, Lines 101-102

Would you discuss why Guanine is not the most abundant in Tagish Lake meteorite? Is it explainable by Fig. 3 as well?

[Reply] Actually, it is not sure why guanine is not the most abundant in the Tagish Lake extract. Also, since the reaction pathway shown in Figure 3 (deleted in the revised version as show later) would not be the only one to yield guanine, it is not easy to relate the lower abundance of guanine with the figure. We can just say Tagish Lake meteorites may have experienced distinct processes with the other two ones which resulted in the lower guanine concentrations, as has been already mentioned in the manuscript.

- Page 10, Lines 186-188

“The absence of pyrimidine in meteorite extracts suggests that the detected pyrimidine nucleobases and their analogs are not formed by the addition of side chains such as methyl (-CH₃) and amino (-NH₂) groups.”

The conclusion sounds a bit strong. Given that the concentrations of the pyrimidine nucleobases are so low, is there possibility that pyrimidine is under detection limit?

[Reply] We did tone down the points a bit and rephrased the sentence as follows: “... suggests that the addition of side chains such as methyl (-CH₃) and amino (-NH₂) groups to pyrimidine may not be the dominant pathway to the formation of the detected pyrimidine nucleobases”.

- Page 12, Lines 230-232

It is suggested that the Tagish Lake and the other two meteorites (Murchison and Murray) may have experienced distinct processes on their parent bodies, since the total concentrations of detected purine molecules in the Tagish Lake and Murray extracts were 9 and 33 ppb, respectively, and those of pyrimidine molecules were 37 and 13 ppb, respectively.

However, the total concentrations of detected purine molecules in the Tagish Lake and (either of) Murchison extracts were mostly consistent, 9 and 11 ppb, respectively, and those of pyrimidine molecules were also consistent, 38 and 37 ppb, respectively. This indicates that differences of the total concentrations of purines and pyrimidines are within the ranges of heterogeneity which the authors discuss.

The authors also mention that the pyrimidines/purines ratios are distinct between Tagish Lake and the other two meteorites. However, the ratio of Murchison #2 is seven times

higher than that of Murchison #1, while the ratio of Tagish Lake is only three times higher than that of Murchison #2. The difference appears to be within the ranges of heterogeneity. Thus, it seems that the total concentrations and their ratios cannot necessarily explain the distinct processes on the meteorite parent bodies.

Rather, Supplementary Figs 8 and 10 are more convincing to explain the difference between Tagish Lake and the other meteorites. I would suggest that Supplementary Fig, 8 should be shown in the main manuscript.

[Reply] Thank you very much for pointing out the important point. Also, we are sorry for the typos in Table 1 for the labelling of Murchison samples. Please replace the expressions “uc” and “mtc” with “#1” and “#2”, respectively (we did so in the revised version). The total purine concentrations in Murchison #1 and #2 are 152 and 11 ppb, respectively. Although the purine concentration in Murchison #2 (11 ppb) is similar to that in Tagish Lake (9 ppb), the total pyrimidine concentration is 2.5 times higher for Tagish Lake (38 ppb) than Murchison #2 (15 ppb). Although the pyrimidine/purine ratio for Murchison #2 is 7 times higher than that for Murchison #1, and that for Tagish Lake is ~3 times higher than Murchison #2, the difference in the absolute value of the pyrimidine/purine ratio is clearly much higher for Tagish Lake than those of two Murchison specimens. Hence, we still expect that the observed difference cannot be explained by sample heterogeneity only. As for Supplementary Figs. 8 and 10, we agree with the reviewer’s opinion. Then we incorporate the Supplementary Fig. 8 in the original version to the main text as Figure 3 in the revised version.

- Supplementary Fig 8

In the relative abundances of imidazole alkyl analogues of Tagish Lake meteorite and the photochemical product, why C8-alkyl imidazole is relatively high abundant?

[Reply] This is an important point that we do not have a clear answer at this time. The molecular formula of the C8-alkyl imidazole is $C_{11}H_{18}N_2$. Although we assume that the detected peaks are all derived from alkylated imidazole analogues, the detected one may not be the case. Instead, it is also likely that this molecule is a structural isomer of C8-alkylated imidazole which does not possess an imidazole structure. In addition, this molecule may easily form by photochemical reactions of interstellar ices, as suggested in the similar trend observed in Supplementary Figure 8. Due to the lack of the corresponding standard reagent, the definite identification was pending in the present status.

- Page 13, Fig. 3

Isn't there a possibility that Xanthine is produced from Guanine by deamination during meteorite parent body alteration?

[Reply] It is likely to occur depending on the availability of water (i.e., aqueous alteration) in the parent body and the temperature at which the meteorite parent body experienced. However, the predominance of guanine in most meteorites implies that the degree of deamination may not be so high.

The authors support the synthetic pathways in Fig. 3 by mentioning that the presence of 5-aminoimidazole-4-carbonitrile was not detected but highly expected. On the other hand, they exclude the synthetic pathway involving the compound (pyrimidine) which they did not detect. This seems to lack consistency in data interpretation and puts their own spin on discussion.

[Reply] In principle, we agree with the reviewer's comment. However, unlike the case of 5-aminoimidazole-4-carbonitrile (Supplementary Figure 13 in the revised version), the presence of pyrimidine in the meteorite extracts is much less expected as shown in the chromatograms below. Nevertheless, as replied above, we have rephrased the sentence about the possible contribution of the pyrimidine route to the formation of pyrimidine nucleobases.

Fig. A. Mass chromatograms of pyrimidine standard reagent and the extract of Murchison #1 at the $m/z = 81.0447$, which corresponds to the m/z of the protonated ion of pyrimidine. The y-axis of the Murchison #1 is almost similar to that of Supplementary Figure 13 in the revised version.

- Page 18, Lines 331-339

It is better to bring this part to the end of the manuscript. I am wondering how effectively such low concentration level (ppt) of nucleobases in meteorites contributed to life's building blocks, in comparison to those produced by the other pathways under different environments, although I am not going to deny your hypothesis...

[Reply] Thank you very much for your comment. We moved the paragraph to the end of the manuscript.

- Page 21, Line 399 Meteorites

If one can evaluate the extent of alteration for the Tagish Lake meteorite used in this study (by comparison with the different lithologies of Tagish Lake meteorite used in Herd et al (2011)), it would be helpful to discuss the relationship between molecular distributions of nucleobases and aqueous alteration.

[Reply] Unfortunately, no information is available about the extent of alteration for the Tagish Lake specimen used in the present study. In that context, we recognize that this is an important discussion, and we are considering these perspectives in our detailed analysis of the Hayabusa2 sample (Tachibana et al., 2021).

Minor comments:

- Page 3, Lines 58-60 “A substantial amount of exogenous meteoritic organics...”

References are needed.

[Reply] Firstly, we rephrased the sentence as follows: “A diverse suite of exogenous meteoritic organics, including nucleobases,...”. We cited here Chyba & Sagan (1992) and Ehrenfreund & Charnley (2000) (refs. #12 & 13 in the original manuscript).

- Page 9, Lines 162-178

Description of future study is lengthy. Would you please shorten this paragraph.

[Reply] We agree. The sentences pointed out were all moved to Supplementary Information.

- Page 9, Line 175

“heavy isotope enrichment in extraterrestrial nucleobases may not be indicative of their extraterrestrial origin.”

-> “heavy C isotope enrichment in extraterrestrial nucleobases may not be indicative of their extraterrestrial origin.” (because the previous study examined only carbon isotope fractionation)

[Reply] Modified as suggested.

- Table 1

Please annotate “uc” and “mtc”.

[Reply] Apologies for the typo. As mentioned above, the expression “uc” and “mtc” correspond to “#1” and “#2”, respectively. Table 1 was modified accordingly. Also, in Table 3, the same expression was used. So it was also modified in the revision.

Reviewer #2 (Remarks to the Author):

The authors describes the identification of pyrimidine nucleobases in organic (formic acid) extracts from the Murchison, Murray, and Tagish Lake meteorites, using high-performance liquid chromatography coupled with electrospray ionization high-resolution mass spectrometry (HPLC/ESI-HRMS). Sample heterogeneity in the organic composition of two Murchison's samples has been highlighted. The novelty of the study is the "first-time" identification of two pyrimidine nucleobases, thymine and cytosine, besides to previously detected uracil, thus completing the panel of canonical pyrimidine nucleobases required for the emergence of primordial DNA and RNA. Others pyrimidine heterocycles and purine nucleobases were detected and quantified, including (for the pyrimidine type) isocytosine, 5-methylcytosine, 6-methylisocytosine and 5-methylisocytosine. In the last part of the manuscript, the detection of pyrimidine nucleobases is associated to photochemical models describing possible synthetic pathway for these compounds.

It is my opinion that the manuscript contains some novelty elements to be published as a communication in Nature, although it requires an extensive reworking and partly revision of the text before the publication (major revision).

[Reply] Thank you very much for your constructive comments. We carefully read your comments and replied to all of them as follows.

First, the overall description of the study is too long and should be reduced significantly. I find the description of the analytical composition of the extracts (first section of the manuscript) the clearest part of the study.

[Reply] We moved a part of paragraph to the Supplementary Information.

Some corrections are required.

They include the revision of the chemical language (and corresponding chemical structures). For example, the definition of pyrimidine nucleobases as "single 6-membered nitrogen (N) heterocyclic ring" is too general. In canonical DNA/RNA pyrimidine nucleobases the nitrogen atoms are always in 1,3 positions and the heterocyclic nucleus of uracil and thymine should be more appropriately reported as

1(H),3(H)-pyrimidin-2,4-dione for etc. The same consideration is valid for purine nucleobases.

[Reply] Thank you very much for the comment. We modified the sentence as follows:
“...pyrimidine nucleobases that consist of a single 6-membered nitrogen (N) heterocyclic ring whose N atoms are always in the 1 and 3 positions and..., and purine nucleobases that consist of 6-membered and 5-membered two ring N-heterocyclic structures whose N atoms are always in the 1, 3, 7, and 9 positions, including...”.

In figures (text and supplementary information) nucleobases are some-time represented as the less thermodynamically stable tautomeric form instead of the more stable one [e.g. N9(H) for adenine etc].

[Reply] The structures of adenine, isoguanine, hypoxanthine and purine were modified appropriately in the main text and SI. Also, for 4(3H)-pyrimidinon, its tautomeric structure, 4-Hydroxypyrimidine, was removed from the main text and SI.

The numbering of atoms in the chemical structures of nucleobases may be useful for a better reading of the text.

[Reply] The numbering was added to the structure of guanine and uracil in Figures 1a and 2a, respectively.

Figure 3 is unnecessary.

[Reply] Deleted.

In the second part, I find the discussion on the possible mechanism of formation of pyrimidine nucleobases very speculative and incomplete. I mean, the sentence “photochemical reactions in the ISM may partly contribute to the presence of nucleobases and related molecules in the Tagish Lake meteorite (pag 19, 357)”, as well as the sentence “The distinct variation pattern of alkylated amines from that of amino acids and imidazoles might imply their different formation processes during the formation of the early solar system (line 33, supplementary information), imply that nucleobases are simply stored in meteorites and does not take into due account recent studies showing that meteorites are catalytically active under different energy conditions and can

both favor the post-ante formation of nucleobases (and other compounds) and lead to their partial or complete degradation (see for example: Chem. Commun., 2019,55, 10563-10566; PNAS J2015, 112, 7109-7110; Chem. Eur. J. 2013, 19, 16916-16922 etc).

In a similar way, the sentence “the validity of the route for the formation of cytosine and uracil starting from cyanoacetylene (HCCCN) (page 20, line 373)” and successive considerations does not take into due account the existence of prebiotic synthetic routes simpler than the use of a complex (astrochemically speaking) three carbon atom precursor such as HCCCN (see: Chem. Eur. J. A 2020, 26, 12075-12080; Chem. Soc. Rev., 2012, 41, 5526–5565 etc).

The next sentence “It is highly probable that, in addition to the pathways proposed above, there should be other routes to the formation of purine and pyrimidine nucleobases present in carbonaceous meteorites” it is inconclusive and does not compensate for the lack of information in the previous section.

I strongly suggest to revise and made more concise the paragraph “Possible formation pathways of nucleobases and their analogs detected in meteorites” limiting it to the essential content and references, reporting on all possible scenarios. Otherwise the paragraph can be completely removed.

[Reply] We reply to all the above four comments on the “Possible formation pathways” section here. We never intend that the reaction route for the synthesis of nucleobases in meteorites is limited to that presented in this section. We showed just an example of potential formation routes proposed so far for readers of Nature Communications. However, as the reviewer pointed out, it was not enough to cite related papers that proposed different pathways toward the formation of nucleobases, many of which have been introduced by the reviewer. Since we expect that the data and scenarios need further experimental validation to be convincing in the revised main text, we focus on the discussion of a possible contribution of photochemical reactions in the interstellar medium to the observed molecular distribution. Accordingly, the title of this section is changed to “**Possible contribution from photochemical reactions in the ISM to the molecular distribution in meteorites**”. Other formation pathways proposed so far, e.g. HCN routes and formamide routes, as well as shock-induced synthesis which was not mentioned in the original version, were described very briefly in the Supplementary Information. As has been mentioned in the text, a full understanding of the formation

pathways of nucleobases in meteorites is beyond the scope of the present study. Also, we do not intend to clarify which routes are more appropriate or not, but just to show examples of previous studies for readers of Nature Communications. Then, we deleted some sentences which sounds like some preference for the HCN-routes in the revised version. We really appreciate the reviewer's suggestion.

In addition, supplementary information should be improved by adding a novel table reporting (for all identified compounds) the intensity and m/z value of the main fragments, highlighting (where necessary) the difference between isomeric structures.

[Reply] We do not think that to show the m/z value and the intensity of the main fragment for all identified compounds are meaningful for readers. Instead, we showed representative mass fragmentation patterns of pyrimidine nucleobases and their structural isomers in the new Supplementary Table 1.

Reviewer #3 (Remarks to the Author):

This is a very interesting paper that reports work investigating the nucleobase contents of carbonaceous chondrites. These compounds are key prebiotic components that could have facilitated the origin of life. The manuscript is well written and gives an good account of previous work. The authors recognised new compounds and identify heterogeneity within samples of the same meteorite.

[Reply] Thank you very much for the very positive comments. We carefully read the comments and modified the manuscript as shown below.

In Line 177, I wondered if the Callaghan et al. experiments did lead to the decomposition of the compounds the authors mention or whether the current protocol was just conservative, which would be fine but could be unpacked briefly in this paper to justify the different procedure.

[Reply] The analysis of nitrogen heterocyclic compounds is not as straightforward as amino acid analysis and requires detailed optimization. We assume the reviewer may point out here the Line 77 (not Line 177) since in the paragraph starting from the Line 77, we mention about the development of analytical techniques. Callahan et al. extracted nucleobases using 95% formic acid at 100 degrees C for 24 h. Although the recovery of this procedure was not reported in Callahan et al., it is likely that some nucleobases could be hydrolyzed, resulting in the deamination (e.g. deamination by hydrolysis of cytosine yields uracil). Rather, what is more critical to yield the difference between their and the present study would be in the other analytical procedures. For example, in their protocol on the solid phase extraction, we confirmed that some of nucleobases were lost during their solid phase extraction procedure (Koga et al. unpublished). In addition, their LC conditions may not be suitable for the detection of tiny amounts of nucleobases in meteorites. In addition, there may be some sample heterogeneity in the content of nucleobases even in the same meteorite as demonstrated in the present study. Such differences in multiple factors may have resulted in the observed difference between the two studies. Nevertheless, we do not prefer to point the finger to their possible weak points in the main text, which we think is not so polite.

The presence of cytosine (Line 156) is interesting and the authors spend some time

discussing its possible indigeneity. In their statements I think they mis-cite reference "37" as "38" (Line 175). In this part of the manuscript they argue that heavy isotope "enrichment in extraterrestrial nucleobases may not be indicative of their extraterrestrial origin". This is hard to understand as written. i) It sounds internally contradictory and ii) Nucleobases on Earth are derived from our highly active biosphere and biological nucleobases do not display the enrichment in heavy isotopes apparent in meteoritic compounds. This part of the paper should be made more clear before publication or it runs the risk of confusing the community.

[Reply] Firstly, as the reviewer pointed out, the citation here was wrong. Thank you very much for pointing it out. As for the argument raised by the reviewer, we do not think it is contradictory. This is because the present biological nucleobases are not synthesized from meteoritic ones. What we would like to mention here is that one general assumption which was prevailing in communities (i.e. heavy isotope enrichment of meteoritic organics is related to their extraterrestrial in origin) is not necessarily correct based on the recent laboratory study (Furukawa et al. 2021). Since the most part of this paragraph including these descriptions were moved to Supplementary Information, we expect that we can avoid the risk the reviewer is worrying about.

Overall, interesting work with implications for parent body processes and origin of life scenarios. It should generate quite a lot of interest from the readership of Nature Communications.

[Reply] Thank you very much again for your very positive comments!

REVIEWER COMMENTS

Reviewer #1 (Remarks to the Author):

Dear Authors

I originally recommended acceptance of the first version of this manuscript. Although there seems to be some comments remained which the authors cannot answer, I can see that the revised manuscript has been improved by all the three reviewers' indications. I would recommend the manuscript for acceptance.

Reviewer #2 (Remarks to the Author):

After the first revision run, the manuscript titled "Identifying the wide diversity of extraterrestrial purine and pyrimidine nucleobases in carbonaceous meteorites" by Yasuhiro Oba et al is (in my opinion) suitable for publication on Nature Communications.

Requests for revision were carefully considered by the authors and the manuscript improved in accordance. My compliments to the authors.